# Expert-generated standard practice elements for evidence-based home visiting programs using a Delphi process

Emily E. Haroz[1]*, Allison Ingalls[1], Karla Decker Sorby[2], Mary Dozier[3], Miranda P. Kaye[4], Michelle Sarche[5], Lauren H. Supplee[6], Daniel J. Whitaker[7], Fiona Grubin[1], Deborah Daro[8]

1 Center for American Indian Health, Department of International Health, Johns Hopkins Bloomberg School of Public Health, Baltimore, MD, United States of America, 2 Minnesota Department of Health, Family Home Visiting, Bemidji, MN, United States of America, 3 Department of Psychology, University of Delaware, Newark, DE, United States of America, 4 Pennsylvania State University, State College, PA, United States of America, 5 Colorado School of Public Health, Centers for American Indian & Alaska Native Health, Aurora, CO, United States of America, 6 William T. Grant Foundation, New York, NY, United States of America, 7 School of Public Health, Georgia State University, Atlanta, GA, United States of America, 8 Chapin Hall, University of Chicago, Chicago, IL, United States of America

* eharoz1@jhu.edu

**Data Availability Statement:** All relevant data are within the paper and its Supporting Information files.

## Abstract

### Background

States, territories, non-profits, and tribes are eligible to obtain federal funding to implement federally endorsed evidence-based home visiting programs. This represents a massive success in translational science, with $400 million a year allocated to these implementation efforts. This legislation also requires that 3% of this annual funding be allocated to tribal entities implementing home visiting in their communities. However, implementing stakeholders face challenges with selecting which program is best for their desired outcomes and context. Moreover, recent reviews have indicated that when implemented in practice and delivered at scale, many evidence-based home visiting programs fail to replicate the retention rates and effects achieved during clinical trials. To inform program implementers and better identify the active ingredients in home visiting programs that drive significant impacts, we aimed to develop an expert derived consensus taxonomy on the elements used in home visiting practice that are essential to priority outcome domains.

### Methods

We convened a panel of 16 experts representing researchers, model representatives, and program implementers using a Delphi approach. We first elicited standard practice elements (SPEs) using open-ended inquiry, then compared these elements to behavior change techniques (BCTs) given their general importance in the field of home visiting; and finally rated their importance to 10 outcome domains.

**Funding:** E.E.H. received funding from the Annie E. Casey Foundation, Inc. for this research and we thank them for their support Grant # GA-2020-B4038): www.aecf.org; however, the findings and conclusions presented in this report are those of the author(s) alone, and do not necessarily reflect the opinions of the Foundation. Author E.E.H. was also supported by the National Institute of Mental Health (nimh.nih.gov) grant number K01MH116335. The funders had no role in study design, data collection and analysis, decision to publish, or preparation of the manuscript.

**Competing interests:** The authors have declared that no competing interests exist.

## Results

Our process identified 48 SPEs derived from the panel, with 83 additional BCTs added based on the literature. Six SPEs, mostly related to home visitor characteristics and skills, were rated essential across all outcome domains. Fifty-three of the 83 BCTs were rated unnecessary across all outcome domains.

## Conclusions

This work represents the first step in a consensus-grounded taxonomy of techniques and strategies necessary for home visiting programs and provides a framework for future hypothesis testing and replication studies.

## Introduction

Home visiting programs focusing on the needs of families and children began as an important strategy in the War on Poverty in the 1960s [1]. Evidence has mounted regarding the effectiveness of home visiting programs to improve maternal and child health, prevent child abuse and neglect, encourage positive parenting, and promote child development [2–5]. The American Academy of Pediatrics® endorses home visiting as a critical strategy to promote child wellbeing and build lifelong health [1]. Currently, $400 million per year through FY2022 has been allocated for home visiting programs through the federal Maternal, Infant, and Early Childhood Home Visiting (MIECHV) and Tribal MIECHV Programs. Many states also offer additional funding for home visiting beyond MIECHV funds because they recognize the importance of home visiting to family and community health and well-being. MIECHV funding is available for states, territories, nonprofit organizations, and tribal nations to provide voluntary home visiting. While there are currently 22 evidence-based models endorsed by the Home Visiting Evidence of Effectiveness federal review, MIECHV grantees may choose 1 or more of 19 models that have implementation support available [1, 6]. Tribal MIECHV grantees may adopt models that are either evidence-based or a promising approach, due to the limited amount of evidence of effectiveness of home visiting programs in tribal communities [7]. Family Spirit® is currently the only home visiting model that meets HHS criteria for evidence of effectiveness.

Home visiting models and their respective interventions are diverse in their theoretical underpinnings and vary in their specific aims, target population, type of home visitors/providers and supervisors, content, schedule of visits, and means of administration. Yet, they are similar in that most provide education, support, and referrals to community services to families living in service areas. In the most comprehensive review of home visiting models, the Home Visiting Evidence of Effectiveness review has examined the available evidence on 50 different home visiting models [8]. Programs range from broad-based models that attempt to change multiple outcomes (e.g., Nurse Family Partnership®, Parents as Teachers®), to specific models focused more narrowly on targeted outcomes such as the Attachment and Biobehavioral Catch-up Intervention [9] or SafeCare® to prevent child neglect and physical abuse [10]. Other home visiting models target certain types of family structures (e.g., adolescent mothers), address the needs of specific cultural or community groups (e.g., American Indian populations), focus only on a certain developmental time period (e.g., Home Instruction for Parents

of Preschoolers [11] or incorporate adjunct services for specific risks such as maternal depression [12].

Given the number of federally endorsed evidence-based home visiting (EBHV) programs (22 at time of publication), stakeholders across local, tribal, and state level organizations face significant challenges identifying and selecting which programs to use based solely on the evidence. Currently, these programs are listed on a central website, HomVEE [13]. The website provides a snapshot of each EBHV model, including details on the populations served and the evidence for certain outcomes. Those interested in implementing home visiting services face the real-world complexities of "evaluating the quality and relevance of competing programs and prioritizing certain outcomes over others in the context of limited time and resources for training and program delivery" [14]. For example, program selection can be complicated by the limited evidence base for the populations the program serves (e.g., American Indian and Alaska Native). Programs may also struggle with how to balance competing priorities. For example, if programs show benefits in one outcome category, but not other outcomes that are of interest, it may be complicated to choose which program to implement in the given service population.

Moreover, several key implementation challenges have been identified as EBHV programs have moved from effectiveness trials to wide scale implementation. These include challenges with client engagement and retention, [4] balancing flexibility and fit during implementation with fidelity to the model that was previously tested [15, 16], and diminished average effect sizes as home visiting programs deliver services at scale [6]. In response to these challenges, program developers are being encouraged by both funders (e.g., MIECHV) and researchers, to unravel their approach and identify, with greater specificity, their key design elements, content, and service delivery strategies. From the research perspective, precision home visiting (PHV) has been championed as a priority to guide this program assessment effort [17].

The precision paradigm primarily emphasizes the importance of identifying the specific behavior change techniques within intervention models and the mechanisms of action by which those techniques promote targeted behaviors and ultimately drive outcomes [17]. While many EBHV models broadly identify target outcomes and the specific changes in participant knowledge, skills and attitudes associated in achieving outcomes, a precision lens asks program developers to apply a more specific and consistent framework in defining their efforts and monitor their implementation and impacts (Fig 1A). While certainly critical to explore in improving program effects, the components of a quality program often extend beyond the direct interaction between a provider and participant. Elements of a program's intervention content, implementation structure, and the required skills and personal characteristics of home visitors also contribute to an intervention's success (Fig 1B).

In an effort to both simplify the decision process for implementers and potentially drive a better understanding of which intervention techniques and delivery methods should be prioritized for future studies, we sought to identify "standard practice elements" (SPEs) across a broad range of strategies that shape the structure of EBHV models. We defined SPEs as the techniques and strategies used in early childhood home visiting as part of the larger intervention. Our approach to identify SPEs was guided by a distillation and matching model, [19, 20] which posits that interventions are conceptualized as composites of individual strategies that can be identified and then matched to client, setting, or other factors that might be relevant for selecting which strategies are most appropriate and when.

Due to the complexities in specifying techniques across a set of models that are highly diverse in their theoretical orientation, discipline, and scope, as a first step in this process, we used a Delphi approach to generate a consensus-grounded taxonomy of techniques and strategies that are used across EBHV models in the United States (US). Briefly, Delphi methods are

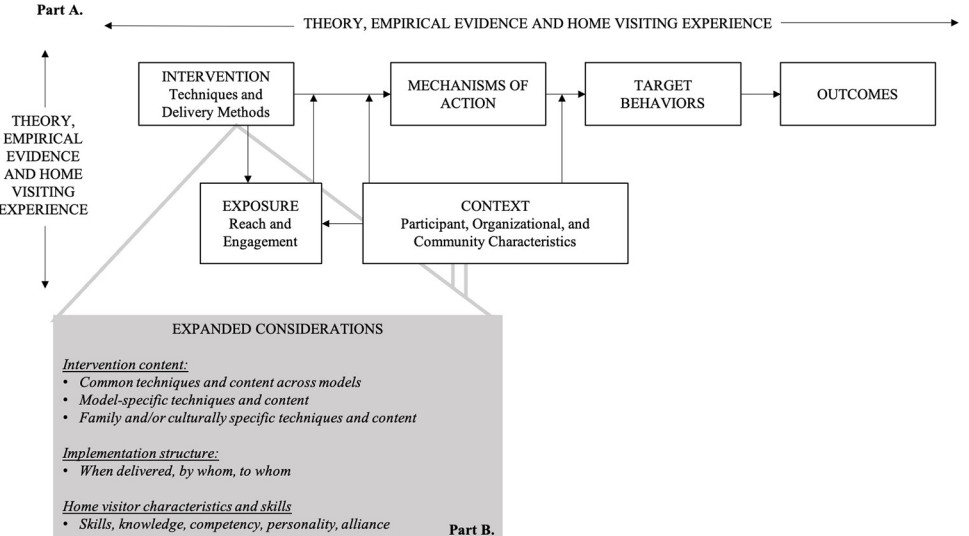

**Fig 1. Expanded version of the home visiting paradigm.** Adapted from Duggan et al., 2021 [18].

structured group communication to identify consensus among a group of experts [21]. Our Delphi approach including recruiting a panel of experts and model developers who met virtually, generated ideas using open-ended techniques, and refined these ideas in an interactive process. The overall goals were to: 1) synthesize the knowledge accumulated over decades of research and practice in home visiting programs to identify SPEs models considered essential to their success; and 2) create a taxonomy of early childhood EBHV SPEs to inform service delivery process, help implementers with identifying and selecting interventions, and inform hypotheses for future PHV research. We specifically focused these efforts on both non-Tribal and Tribal Home Visiting (THVs) contexts, because of the overlapping funding streams, but different processing for selecting EBHVs, and with respect for Tribes as sovereign nations within the US.

## Methods

The Delphi process [22] was conducted over a nine-month period beginning in May 2020 and ending in January 2021 (Fig 2).

### Participants

Study team members employed purposive sampling [23] to identify an expert panel including members representing early childhood home visiting researchers, allied experts from home visiting models, and leaders in THV development and implementation [23]. Participants came from diverse geographic areas of the US and represented several sectors of the home visiting field including model developers, implementers, and researchers. At the end of March 2020, the study team emailed an invitation to participate to 22 home visiting experts. Three potential panel members were unable to participate due to other commitments. Three model developer representatives never responded to the invitation. All invited leaders in THV agreed to participate in the study. Ultimately, there were 16 members of the expert panel, each of them participating in some capacity during all three rounds of the Delphi process. Panel members held multiple rolls, including n = 10 (63%) who identified as a researcher, n = 7 (43%) who identified as a model representative, and n = 7 (43%) who identified as a tribal stakeholder.

**Process**

**Taxonomy of Standard Practice Elements (SPEs)**

| Process | Taxonomy |
|---|---|
| Formation of expert panel and overview of elicitation process (March-May 2020) | |
| **Round 1** — Survey #1 (May 2020) Free listing of SPEs (n=16 respondents) | **Initial list with deduplication (n=68, includes n=10 tribal home visiting (THV) SPEs)** |
| Video Conference #1 (June-August 2020) Initial revision of panel-generated list of SPEs; group consensus on broad category names and definitions | **Revised list (n=69, includes n=11 THV SPEs) and 5 broad category names, definitions, and examples** |
| **Round 2** — Survey #2 (September 2020) Categorization of SPEs into broad categories (n=14 respondents) | **Revised list (n=62, includes 11 tribal home visiting SPEs) grouped into broad categories (n=5)** |
| Video Conference #2 (October 2020) Revision of standard practice elements | **Revised list (n=61, includes 10 tribal home visiting SPEs)** |
| Survey #3 (October 2020) Clarification of select SPEs and revision of taxonomy (n=16) | **Revised list (n=45, includes 5 tribal home visiting SPEs)** |
| **Round 3** — Matching SPEs to Michie's Behavior Change Technique (BCT) Taxonomy | **Revised list of SPEs (n=38), SPEs/BCTs that overlapped (n=10); added unique BCTs (n=83) for a total of N=131 SPEs/BCTs** |
| Survey #4 (December 2020-January 2021 Prioritization of SPEs and BCTs to selected outcome domains (n=16) | **Final taxonomy (N=75 SPEs & BCTs) prioritized to selected EBHV outcome domains & analyzed as heatmaps** |

**Fig 2. Delphi process.** Gray boxes represent preparatory steps by the research team; white rectangular boxes outlined in black illustrate core events for collecting input and decision-making; and bolded text represents the development of a taxonomy of evidence-based home visiting standard practice elements. The members of the research are not included in the numbers of respondents to surveys.

## Procedures

Our approach was guided by the Delphi method, which was developed by the RAND Corporation in the mid-20th century as a way for researchers to reliably build consensus of a group of experts [24, 25]. Because this study took place during the COVID-19 pandemic, we the Delphi approach with the three rounds of questionnaires and discussion carried out via a series of Qualtrics surveys [26] and Zoom video conferences, [27] coupled with email communication as necessary. Since the expert panel was only asked questions about home visiting in general and no personal information was solicited, this process was exempt from oversight by the Institutional Review Board [28]. As such, while participants did not provide formal informed consent, they did agree to participate prior to the process as part of the virtual convenings and surveys.

At the start of the expert elicitation process, participants were divided into two panels: one for researchers and model developers, and one for leaders in THV. As such, two project launch meetings were conducted via Zoom video conference in May 2020 to provide expert panel members with an overview of the project prior to launching into Round 1 of the Delphi process.

**Round 1.** Both expert panels received the same survey. In this survey, free lists were used to elicit SPEs in EBHV. Free listing was selected as a feasible approach that would preserve open-ended inquiry. Participants were provided with a definition of SPEs—"the techniques and strategies used in early childhood home visiting as part of a larger intervention"—and were asked to list out all elements they considered important in early childhood home visiting. There was room to enter up to 20 individual SPEs, in addition to an open-text field for additional entries. Participants were then asked to identify which free-listed SPEs are critical to home visiting programs that serve tribal communities. Lastly, participants were asked to list any additional SPEs specific to THV but not already mentioned previously. There was space for up to five individual tribal SPEs, in addition to an open-text field for additional entries. Data collection lasted for 3 weeks, concluding when the target sample size ($n = 16$) was reached. To view the full first survey, refer to S1 File.

*Analysis of Round 1*. Responses to the first survey were exported and combined into one Excel file with two tabs labeled "general elements" and "tribal elements." Each tab in the Excel file had four columns: element (i.e., standard practice element label), frequency (i.e., count of respondents who free-listed the same or similar element), respondent initials (so we could follow-up if needed), and other similar descriptors (so we could see how each element was coded). Study team members read through each respondent's free-listed SPEs and decided whether each was distinct or could be combined with another element already listed in the Excel file. For example, the element "linkage to services" was listed by 12 respondents using different descriptors such as "connecting families to resources," "making a referral to appropriate community service," and "resource connection," among others. If it was unclear that listed elements were referring to the same thing, the study team kept them separate.

Two additional video conferences were held with each of the expert panels in June 2020 to further clarify and refine the list generated through the initial study. Study team members guided these group discussions to identify commonalities, clarify meaning, and solicit additional thoughts. In addition, during these meetings each panel independently concluded that there needed to be a broader level of categorization of SPEs to capture the multi-level hierarchy of home visiting (e.g., what are elements delivered during the visit, what are elements for home visitors training and background, etc.). As a result, via email feedback, panel members were asked to build consensus on broad category names and definitions to be used in future Qualtrics surveys.

**Round 2.** For Round 2, the expert panels were combined, and they remained combined through Round 3. Thus, all email communication, subsequent surveys, and video conferences were held with the entire group of experts. In September 2020, expert panel members were asked to complete a second Qualtrics survey. In this survey, all previously elicited SPEs were presented to participants, and they were asked to group these into the broad categories defined in Round 1. Because each SPE was listed as a multi-select question type, they were able to be grouped into multiple broad categories. At the end of the survey, participants had the option to add additional SPEs that were not already included in the master list. Data collection lasted for 3 weeks, and responses were collected for n = 14 expert panel members. The full survey is available in S2 File.

The second phase of Round 2 included a second video conference held with all expert panel members at the beginning of October 2020. The study team presented the results of the second

survey and asked follow-up questions to further collapse or clarify SPEs. This included re-wording of some elements and the addition of elements to better capture specific practice activities. These additions were done through consensus processes across panel members. Due to several elements that continued to need clarity, expert panel members were asked to complete a third, brief Qualtrics survey to help finalize the list of SPEs. Participants were also asked for further input (i.e., "Is this definitely a standard practice element in home visiting?") on free-listed elements that were initially provided by only one panel member in the first Qualtrics survey. Data collection lasted for a little over 1 week, concluding when the target sample size (n = 16) was reached. To view the full survey, refer to S3 File.

*Data analysis for Round 2.* Responses to the first survey in Round 2 were exported, and a study team member tallied the number of responses for each SPE under each broad category. The full study team then met to make decisions about final SPE-broad category matching. A rule was created that for any SPEs that were added to a broad category by most respondents (greater than 50%), it would be considered assigned to that category. For example, the SPE "teaching goal setting skills to parents" was voted by nine expert panel members as belonging to the broad category "home visiting content." Since that is more than half of the 14 total respondents to that survey, it was coded in that category for the results. Ties were discussed as a study team. Responses for the second survey in Round 2 were similarly exported and analyzed with SPEs only included in the round 2 list if greater than 50% of respondents endorsed their inclusion.

**Round 3.** To align with other efforts in the PHV field, [17, 29] and to keep consistent with several home visiting program approaches aimed at changing parental behavior, at the beginning of this final round of the Delphi process, study team members engaged in an internal process to match expert-generated SPEs to the Behavior Change Technique Taxonomy [30]. The Behavior Change Technique Taxonomy was developed by researchers in the United Kingdom through a Delphi process in order to provide structure for reporting on behavior change interventions. Behavior change techniques (BCT) are strategies that help an individual change their behavior to help achieve better health. Identifying and understanding the mechanisms of BCTs are the current focus of most PHV research. Our approach to defining SPEs was intentionally broader–focusing not just on individual change, but techniques and strategies used broadly in early childhood home visiting programs to effect change. First, a full list of SPEs and BCTs was created, along with definitions and examples for each item in the list. The Behavior Change Technique Taxonomy had published definitions and examples, but the study team created definitions and examples for the panel generated SPEs based on our panel discussion notes and from experience in the home visiting field.

The list of SPEs and BCTs was input into a Qualtrics survey using the "Pick, Group, and Rank" question type to facilitate independent coding between two internal coders from the study team. Each coder independently filled out the survey by dragging and dropping panel generated SPEs into matching BCTs. The two coders and Principal Investigator discussed the results of the survey and came to consensus about any disagreements in the coding. Afterward, a final list of all SPEs and BCTs was created.

For the final Round 3 of Delphi surveys, panel members completed a Qualtrics survey to prioritize SPEs and BCTs to outcome domains in evidence-based home visiting (S4 File). Outcome domains selected for this project were adapted from the Home Visiting Evidence of Effectiveness review and the Pew Home Visiting Data for Performance Initiative [31, 32] with two additional tribal outcome domains that were of interest to the study team. Domains included: 1) promotion of healthy physical child development (e.g., healthy eating, breastfeeding); 2) promotion of social-emotional learning; 3) improving cognitive development (e.g., language development); 4) linkages and coordination of referrals for other community

resources and supports; 5) reductions in maternal distress (e.g., depression, anxiety, stress); 6) reductions in substance use; 7) promotion of positive parenting practices; and 8) reductions in child maltreatment. Domains specific to THV included: 9) reductions in tribally related health disparities (e.g., Type 2 Diabetes, Mental Health); and 10) promotion of connection to culture. Because the list of SPEs and BCTs was so long, each expert panel member was randomly assigned to complete the matching survey for only four outcome domains. Ratings were chosen based on methods used in a previous study by McLeod et al. [33] and included: 0 = "Not necessary," 1 = "Useful, but not essential," and 2 = "Essential" to achieve the outcome. Expert panel members rated each SPE and BCTs according to their importance in achieving the outcome domain. Data collection lasted for 7 weeks, concluding when the target sample size (n = 16) was reached. To view the full final survey, refer to S5 File.

*Data analysis for Round 3*. Data were exported and combined in Excel for each outcome domain. For each element, frequency of ratings (e.g., Not necessary, Useful, Essential) and average ratings were tallied for each outcome domain. Average ratings were used as an indicator for strength of the relationship between the element and the outcome which were analyzed as heatmaps to help with visualization of the results.

### Final taxonomy

Final taxonomies of SPEs and BCTs were developed for each outcome domain. This was done by retaining any SPE or BCT that was rated as "Essential" to that outcome domain by 50% or more of respondents.

## Results

### Round 1

In total, there were 58 SPEs free listed by expert panelists in the first Qualtrics survey, along with 10 unique THV SPEs, creating a total of 68 free-listed EBHV SPEs. Of these, 51 general SPEs were also thought to be critical to THV. After the Zoom video conference to discuss survey results, the list was revised to comprise a new total of 69 SPEs, including 11 THV SPEs. Group consensus was reached on n = 5 broad category names, definitions, and examples (See Table 1).

### Round 2

After the second Qualtrics survey, the list of SPEs included *n* = 62 individual elements (51 general, and 11 unique to THV) categorized into n = 5 broad categories. Results from the group discussion and third Qualtrics survey lead to a reduction in total number of SPEs (n = 45, including 5 unique to THV). The frequencies by broad category name for the final round 2 survey can be found in Table 1. The category with the most SPEs assigned to it was home visiting process/ delivery (n = 24), followed by program implementation (n = 15) and home visitor personal characteristics (n = 11). Model philosophy and home visiting content each had six SPEs assigned to them.

### Round 3

After internal matching of SPEs to BCTs, the draft taxonomy grew to 38 SPEs, 10 SPEs/BCTs that overlapped, and 83 BCTs. Expert panel members used this list to rate each element by importance to achieving each outcome domain. Fig 3 presents a heatmap that displays the average rating of 28 of the elements by outcome domain, with darker shade indicating higher average rating across participants. The heatmaps for the remainder of the elements can be

Table 1. Frequency of SPEs* by category domains.

| Category Domain[a] | Consensus Definition | Number of Round 2 SPEs[b] | Examples of SPEs |
|---|---|---|---|
| Model philosophy | The tenets of an evidence-based home visiting program that drive the other components of home visiting, including a model's theory of change and cultural lifeways. | 6 | Model is based on a parenting framework; Home visitor understands, affirms, and respects cultural identity of clients (THV)* |
| Program implementation | Strategies, techniques, structures, and processes (e.g. program design) at the model/organizational/site level that relate to ensuring successful delivery of the EBHV, including buy-in (community, agency, home visitor), staff training, supervision, fidelity, funding, and payment structures. | 15 | Providing clients with linkage to services; Program strengthening of service coordination; Culturally attuned and responsive approach with all staff training, strategies, materials (THV) |
| Home visiting content | The content (i.e. the "what") that is conveyed by home visitors to their clients during service delivery. | 6 | Teaching relaxation/ self-regulation skills to parents; Teaching goal setting skills to parents |
| Home visiting process/ delivery | The strategies and techniques home visitors use during service delivery with their clients. | 24 | Child assessment and screening; Active listening; Role play/ coaching; Home visitor shares resources in client's Native language (THV) |
| Home visitor personal characteristics | Home visitor characteristics that may contribute to improvement in client outcomes but that aren't typically specified in a model's theoretical/ conceptual framework or their content. These are aspects of the home visitor that are not an explicit part of the model. | 11 | Home visitor flexibility/ adaptability; Reliable home visitor; Home visitor sense of humor |

* SPE = standard practice element; THV = tribal home visiting

a Categories are not listed in order of priority, and they are not listed in a hierarchy.

b Not mutually exclusive

found in Supporting information. Average ratings range from 0 to 2.0. The highest rated element across all outcome domains was "relationship building" while nine elements were rated zero across all outcome domains. All the elements that were rated zero across all outcome domains were from the BCT.

## Final taxonomy

Using a cut-off point of 50% or more respondents rating the element as 'Essential' to classify the element as *essential* to changing the specified outcomes, the number of SPEs and BCTs classified as essential by domain can be found in Table 2. Maternal distress included the greatest number of SPEs and BCTs with 51 elements (34 SPEs and 17 BCTs) classified as essential. No BCTs were classified as essential to increasing referrals as all 22 elements came from the expert generated SPEs.

Altogether, six SPEs were rated essential across all outcome domains while 54 elements were not rated as essential across any of the outcome domains (Table 3). Five out of the six SPEs rated as essential for all outcome domains represented key personal characteristics and skills of home visitors (Table 3). The one additional SPE not related to home visitor personal characteristics was: "Culturally attuned and responsive approach with all staff training, strategies, materials" (Table 3). Content based home visiting techniques that emerged as highly rated across 90% of outcomes included 1) Providing clients with linkages to services; and 2) Maternal risk screening and assessment (Table 3). Across all ten outcome domains of home visiting, 75 SPEs were classified as essential to achieve at least one outcome, but only fifteen elements were classified as essential across at least nine of the outcome domains (Table 3).

Fig 3 shows a heatmap for 28 SPEs rated across all outcome domains and a sum of their ratings across all domains except for the two tribal home visiting specific outcomes. Remaining heatmap figures can be found in Supporting information. Again, many of the most highly

| Element | Category* | Child physical development | Child social emotional learning | Child cognitive development | Increased referrals | Maternal distress | Parental substance use | Positive parenting | Child maltreatment | Tribal health disparities | Connection to culture | Sum of average scores** |
|---|---|---|---|---|---|---|---|---|---|---|---|---|
| Creating an action plan based on child screenings | SP | 2.00 | 2.00 | 1.71 | 1.20 | 0.86 | 0.00 | 1.56 | 1.33 | 1.33 | 0.29 | 10.66 |
| Reflect on strategies to support results from caregiver screenings | SP | 1.67 | 0.67 | 0.75 | 1.20 | 1.14 | 1.60 | 1.11 | 1.00 | 1.33 | 0.29 | 8.96 |
| Home visitor facilitates client connection to cultural and/or spiritual resources | SP | 1.00 | 0.67 | 0.75 | 0.80 | 0.57 | 0.80 | 0.67 | 0.33 | 1.33 | 1.50 | 5.19 |
| Motivational interviewing | SP | 0.33 | 0.67 | 0.00 | 0.80 | 0.86 | 1.20 | 0.67 | 1.00 |  | 0.25 | 5.52 |
| Teaching coping skills to parents | SP | 1.00 | 1.67 | 0.00 | 0.40 | 1.71 | 1.20 | 1.33 | 1.67 |  | 0.29 | 8.98 |
| Teaching relaxation/ self-regulation skills to parents | SP | 0.67 | 1.00 | 0.25 | 0.00 | 1.43 | 0.80 | 0.89 | 1.00 | 0.22 | 0.00 | 6.07 |
| Home visitor shares resources in client's Native language | SP | 0.67 | 1.33 | 1.14 | 0.80 | 0.57 | 0.80 | 0.67 | 0.67 | 0.89 | 1.00 | 6.36 |
| Providing clients with linkage to services | SP | 1.67 | 1.33 | 1.14 | 2.00 | 1.71 | 1.60 | 0.89 | 1.33 | 1.33 | 1.25 | 11.68 |
| Child assessment and screening | SP | 2.00 | 1.67 | 2.00 | 1.60 | 0.57 | 0.00 | 1.33 | 0.67 | 1.56 | 0.25 | 9.84 |
| Maternal risk assessment and screening | SP | 1.00 | 1.67 | 1.43 | 1.60 | 1.43 | 2.00 | 1.56 | 1.67 | 1.56 | 0.25 | 12.35 |
| Reflective supervision | SP | 1.00 | 1.00 | 0.57 | 0.00 | 0.29 | 0.80 | 0.67 | 0.67 | 0.22 | 0.25 | 4.37 |
| Professional development | SP | 1.33 | 1.33 | 1.43 | 0.40 | 0.86 | 0.80 | 0.89 | 0.67 | 0.89 | 1.00 | 7.71 |
| Proper workloads of staff/ supervisors | SP | 2.00 | 2.00 | 1.43 | 1.60 | 2.00 | 0.80 | 1.78 | 1.33 | 1.33 | 1.25 | 12.94 |
| Criteria for staff selection are appropriate for population served | SP | 1.67 | 1.33 | 1.43 | 1.20 | 1.14 | 0.80 | 1.56 | 1.00 | 1.56 | 1.00 | 10.13 |
| Home visitor flexibility/ adaptability | SP | 1.33 | 1.33 | 1.14 | 1.20 | 1.43 | 0.80 | 1.11 | 1.00 | 1.33 | 1.50 | 9.35 |
| Home visitor sense of humor | SP | 0.00 | 0.00 | 0.00 | 0.00 | 0.29 | 0.00 | 0.22 | 0.00 | 0.22 | 0.25 | 0.51 |
| Reliable home visitor | SP | 2.00 | 2.00 | 1.14 | 1.20 | 2.00 | 0.80 | 2.00 | 1.33 | 1.78 | 1.50 | 12.48 |
| Active listening | SP | 1.67 | 1.67 | 0.86 | 1.60 | 1.71 | 1.20 | 2.00 | 1.33 | 1.78 | 1.50 | 12.04 |
| Relationship building | SP | 2.00 | 2.00 | 1.43 | 1.60 | 2.00 | 1.20 | 2.00 | 1.67 | 1.78 | 1.50 | 13.90 |
| Responsiveness and sensitivity | SP | 1.67 | 2.00 | 1.43 | 1.60 | 2.00 | 1.20 | 2.00 | 1.67 | 2.00 | 1.50 | 13.56 |
| Home visitor demonstrates cultural humility | SP | 1.67 | 1.67 | 1.71 | 1.60 | 1.71 | 1.20 | 1.56 | 1.33 | 1.78 | 2.00 | 12.45 |
| Empathetic communication | SP | 1.67 | 1.67 | 1.00 | 0.80 | 1.71 | 1.67 | 1.56 | 1.67 | 1.33 | 1.50 | 11.74 |
| Home visitor discipline regarding boundaries and limits of their role | SP | 0.67 | 1.33 | 1.00 | 0.80 | 1.71 | 1.33 | 0.89 | 1.33 | 0.67 | 1.00 | 9.07 |
| Program trains staff on the prevalence, causes, and consequences of trauma | SP | 0.67 | 1.00 | 0.25 | 0.00 | 0.86 | 0.13 | 0.67 | 0.67 | 1.11 | 0.75 | 4.44 |
| Program strengthening of service coordination | SP | 1.67 | 0.67 | 0.75 | 1.60 | 1.14 | 0.67 | 0.67 | 1.00 | 1.11 | 1.00 | 8.16 |
| Home visitor content mastery | SP | 1.67 | 1.20 | 1.50 | 1.20 | 2.00 | 0.67 | 1.11 | 1.67 | 1.33 | 1.25 | 11.01 |
| Culture of quality for implementing program | SP | 1.67 | 1.33 | 1.00 | 0.80 | 1.71 | 0.67 | 1.33 | 1.00 | 1.11 | 1.00 | 9.51 |
| Organization/ program collaboration and outreach across the community | SP | 1.67 | 0.67 | 0.75 | 2.00 | 1.14 | 0.67 | 0.67 | 1.00 | 1.33 | 1.50 | 8.56 |

**Fig 3. Heatmap of SPEs and BCTs and their relative importance across outcome domains.**

**Table 2. SPEs and behavioral change techniques classified by 50% or more respondents as essential by domain.**

| Outcome Domain | Number of SPEs + Behavior Change Techniques | Number of unique SPEs | Number of unique Behavior Change Techniques | Number of non-essential SPEs | Number of non-essential Behavior Change Techniques |
|---|---|---|---|---|---|
| Child physical Development | 47 | 29 | 11 | 10 | 72 |
| Child social emotional learning | 40 | 37 | 3 | 9 | 80 |
| Child cognitive development | 35 | 27 | 8 | 19 | 75 |
| Increased referrals | 22 | 22 | 0 | 24 | 83 |
| Maternal distress | 51 | 34 | 17 | 12 | 66 |
| Parental substance use | 30 | 20 | 10 | 26 | 73 |
| Positive parenting | 37 | 29 | 8 | 17 | 75 |
| Prevention of maltreatment | 45 | 35 | 10 | 11 | 73 |
| Tribal health disparities | 33 | 27 | 6 | 19 | 77 |
| Connection to culture | 25 | 23 | 2 | 23 | 81 |

**Table 3. Elements classified as essential across 90% or more of the outcome domains.**

| Element | Child physical development | Child social emotional learning | Child cognitive development | Increased referrals | Maternal distress | Parental substance use | Positive parenting | Child maltreatment | Tribal health disparities | Connection to culture |
|---|---|---|---|---|---|---|---|---|---|---|
| 1. Relationship building | √ | √ | √ | √ | √ | √ | √ | √ | √ | √ |
| 2. Responsiveness and sensitivity | √ | √ | √ | √ | √ | √ | √ | √ | √ | √ |
| 3. Home visitor demonstrates cultural humility | √ | √ | √ | √ | √ | √ | √ | √ | √ | √ |
| 4. Home visitor adaptability with respect to setting and participation | √ | √ | √ | √ | √ | √ | √ | √ | √ | √ |
| 5. Home visitor understands, affirms, and respects cultural identity of clients | √ | √ | √ | √ | √ | √ | √ | √ | √ | √ |
| 6. Culturally attuned and responsive approach with all staff training, strategies, materials | √ | √ | √ | √ | √ | √ | √ | √ | √ | √ |
| 7. Providing clients with linkage to services | √ | √ | √ | √ | √ | √ | X | √ | √ | √ |
| 8. Maternal risk assessment and screening | √ | √ | √ | √ | √ | √ | √ | √ | √ | X |
| 9. Proper workloads of staff/ supervisors | √ | √ | √ | √ | √ | X | √ | √ | √ | √ |
| 10. Criteria for staff selection are appropriate for population served | √ | √ | √ | √ | √ | X | √ | √ | √ | √ |
| 11. Home visitor flexibility/ adaptability | √ | √ | √ | √ | √ | X | √ | √ | √ | √ |
| 12. Reliable home visitor | √ | √ | √ | √ | √ | X | √ | √ | √ | √ |
| 13. Active listening | √ | √ | X | √ | √ | √ | √ | √ | √ | √ |
| 14. Empathetic communication | √ | √ | √ | X | √ | √ | √ | √ | √ | √ |
| 15. Home visitor content mastery | √ | √ | √ | √ | √ | X | √ | √ | √ | √ |
| 16. Culturally informed knowledge of the home visitor | X | √ | √ | √ | √ | √ | √ | √ | √ | √ |

rated SPEs and BCTs relate to home visitor personal characteristics or non-specific skills such as "Relationship building." Additional content based home visiting techniques that emerged as highly rated included: 1) Creating action plans based on child screenings; and 2) Teaching problem solving skills (Figs 3 and S1). Home visitor content mastery was also highly rated despite relatively lower content based home visiting techniques identified as essential across outcomes. This may be because some models include specific content related to a limited set of

outcomes (e.g., SafeCare focuses content only on prevention of child maltreatment). All but one of the 54 elements that never achieved an essential rating by more than 50% or respondents were from the behavior change taxonomy, except for "Home visitor sense of humor," which was an SPE.

## Discussion

The development of a consensus-grounded taxonomy of techniques and strategies used across EBHV programs sets the foundation for necessary next steps in the implementation and scaling EBHV programs. First, the home visiting research field has begun to explore the potential benefits of conducting PHV studies to strengthen programs and outcomes for families. A necessary part of PHV research is to identify the core techniques and strategies that may drive outcomes. Without consensus on which techniques and strategies are most promising to investigate, the PHV research may end up with a proliferation of studies with an inability to replicate findings and draw cohesive conclusions. Second, identifying the core techniques and strategies has the potential to help those identifying and selecting EBHV interventions that best meet their communities' and families' needs. For example, our findings suggest that, regardless of the home visiting model used, focusing on the qualifications and skills of home visitors is critical to maximizing outcomes. Moreover, programs can use Fig 3 to identify key features of home visiting programs that are considered important to include in programming for the selected outcome(s).

Our results indicated that across 10 common outcome domains of home visiting, 75 SPEs were classified as essential for home visiting models to include to achieve at least one outcome. While we included ratings for BCTs, very few BCTs were identified as essential to achieving outcomes in home visiting. The highest rated BCTs overlapped with expert generated SPEs and included: "Information sharing (by home visitor to client)" and "Teaching problem solving skills to parents." Moreover, 10 of the SPEs identified as essential across almost all outcome domains related to home visitor personal characteristics and non-specific trained skills rather than content-based techniques or delivery methods. These results suggest that while some content-based home visiting techniques and delivery methods are essential to achieving outcomes, the process by which content is delivered and who it is delivered by may drive change in priority outcomes. This finding is consistent with findings in the psychotherapy field [34]. Future work is needed to identify important characteristics of home visitors, and training approaches for these home visitors in the skills that are hypothesized to drive effective programming.

From the outset of this process, the expert panel members emphasized a need to think about how SPEs fit into broader based categories of program implementation and effectiveness. The five broad based categories that were identified included: 1) Model philosophy; 2) Home visiting content; 3) Home visitor personal characteristics; 4) Home visiting process/delivery; and 5) Program implementation. These broad categories are consistent with the home visiting precision paradigm and point to a need for further studies that focus not only on the active ingredients of a particular model, but also consideration for how implementation science principles impact the context of home visiting. The context of the home visiting program plays a key role in affecting outcomes. Implementation outcomes are known in the literature to affect participant outcomes, particularly prevention of child maltreatment [35]. There is a need to understand what types and how implementation strategies can contribute to addressing the current challenges in the home visiting field.

Despite behavior change being considered a key activity in home visiting programs through home visitor support of parents to engage in healthy behavior and positive parenting, [36] few of the existing behavior change strategies overlapped with the expert-developed list of SPEs

and almost none were considered essential to achieving home visiting outcomes across any outcome. This is notable, particularly in the context of the PHV field. Much of existing PHV work has been focused on a paradigm that behavior change is the ultimate goal in most home visiting programs. Therefore, the existing Behavior Change Technique Taxonomy should provide a rather exhaustive list of the elements necessary to achieve this change [30]. However, experts who participated in this Delphi process clearly did not agree that BCTs drive positive change in home visiting. Changing behavior may be more complicated than specific techniques and may rather focus on a combination of characteristics, techniques, and implementation context that ultimately translates to effective programming. While our investigation focused on ten outcomes of home visiting programs, it is possible that BCTs may be particularly relevant to certain types of outcomes such as changing unhealthy eating habits [37–39] which were not captured in our approach. Our findings also point to the need to broaden our consideration of active ingredients to test in PHV to investigate and include the SPEs identified here.

Due to the unique contexts and funding streams for THV efforts, and the need to prioritize populations who face significant health disparities, our process focused explicitly on identifying SPEs for eight standard outcomes with the addition of two outcomes that were determined to be particularly relevant for THV contexts: 1) Promoting a connection to culture; and 2) Reducing tribally relevant health disparities. The SPEs that focused specifically on the home visitor's cultural knowledge, awareness, humility, and sensitivity were considered particularly important for these outcomes. This is consistent with the broader prevention literature in tribal communities, [40] but also relevant when working on prevention efforts in any community [41].

Bridging the research-to-practice gap by summarizing EBHV models' "standard practice elements" and, ultimately, identifying those that are common or overlapping may be particularly useful for several reasons. First, it may better inform practice because it begins to build a framework for stakeholders that would more easily enable them to personalize their programming to ensure all desired elements are available. While we did not perform a meta-analysis, our approach may help implementers identify where models overlap or are distinct from one another and use this information to select or tailor their own approaches. Second, our approach does not provide information about the impact of each SPE on outcomes, but instead provides a starting point for researchers to identify which elements might be tested further in their pursuit of identification of empirically supported active ingredients. And third, our hope is to contribute to a shared vision of improving the health and wellbeing of families and communities.

Identifying common practices across EBHV models can ultimately lead to more structured tailoring of interventions to better fit the unique needs of participating families. The SPEs may provide a menu of choices from which to pick to personalize a home visiting program for each family and their unique needs. For example, if a mother scores highly on mental health screening, one of the major goals of their home visiting program might be teach coping skills within the program and to refer and connect them for services. Similarly, if the home visitor is concerned about the child's development, administering screening, and developing an action plan may be a priority. These practice elements could also be combined if there are multiple, competing priorities. While more research is needed to empirically prove the effectiveness of different SPEs for different outcomes, for now, stakeholders, including program administrators and home visitors, could review home visiting curriculums to identify priority outcomes and ensure the SPEs for those outcomes are included in their current practice.

### Limitations

There are limitations to consider when interpreting the results of this study. First, the evidence base for each SPE was not considered because the purpose was to identify a wide range of possible elements that could be considered. Second, while we attempted to link these elements to a relevant theory in the behavior change literature, other theories or conceptual frameworks may also be considered and might further strengthen this approach. Third, while we attempted to include experts representing a variety of stakeholder perspectives, our sampling is not fully representative of the home visiting field. For example, we did not include the voice of home visitors or clients, who may place value on other elements. The list of combined SPEs and BCTs and number of outcomes was large, preventing coding by all participants. Further refinement of the final taxonomy and by a wider audience may be helpful. Finally, we did not evaluate how well or to what extent existing evidence-based home visiting models incorporate the SPEs identified. Exploring to what extent model include these SPEs would be an important future research direction.

## Conclusions

This project represents a preliminary step in identifying SPEs that cut across EBHV programs. By starting with open-ended processes (i.e., free listing) we were able to elicit a wide range of potential SPEs based on extensive expertise in the field. This type of inquiry is especially helpful in generating hypotheses. The field of PHV is dedicated to identifying active ingredients but knowing where to start in terms of which active ingredients to investigate and why is a critical first step in any scientific endeavor. In the short term, models can use the information included in this manuscript to ensure their programs include the essential elements. Next steps for further research might involve refinement of this initial taxonomy with a broader range of stakeholders and ultimately coding of actual program materials to identify common practice elements across models. The move towards identifying SPEs across models holds potential to overcome existing implementation challenges and ultimately strengthen impact on children's and parents' health, increase home visitor efficiency, and lower program cost while increasing economic and societal benefit.

## Supporting information

**S1 File. Expert panel survey 1—free listing.**
(PDF)

**S2 File. Expert panel survey 2—categorizing SPEs.**
(PDF)

**S3 File. Expert panel survey 3 –further input on SPEs.**
(PDF)

**S4 File. Internal team survey matching to BCTs.**
(PDF)

**S5 File. Expert panel survey 4—prioritizing SPE.**
(PDF)

**S6 File. Data file from Delphi round 1.**
(XLSX)

**S7 File. Data file from Delphi round 2.**
(XLSX)

**S8 File. Data file from Delphi round 3.**
(XLSX)

**S1 Fig.**
(TIF)

**S2 Fig.**
(TIF)

**S3 Fig.**
(TIF)

**S4 Fig.**
(TIF)

## Acknowledgments

We continue to be immensely honored to work on behalf of all the participating families in home visiting across the country. We gratefully acknowledge expert panel members who are co-authors on this manuscript: DD, KDS, MD, MPK, MS, LHS, and DW. In addition, we appreciate all contributions made by expert panel members who do not meet the criteria for authorship but who nonetheless provided invaluable input: Dr. Allison Barlow, Johns Hopkins Bloomberg School of Public Health, Center for American Indian Health; Jill Filene, James Bell Associates; Dr. Kathryn Harding, Prevent Child Abuse America; Dr. Christa Haring Biel, Children's Equity Coalition; Crystal Kee, Zero to Three; Allison Kemner, Parents as Teachers; Lisa Martin, Inter-Tribal Council of Michigan, Inc.; Dr. Nancy Rumbaugh Whitesell, Colorado School of Public Health, Centers for American Indian & Alaska Native Health; and Dr. John Walkup, Ann & Robert H. Lurie Children's Hospital of Chicago, Department of Psychiatry and Behavioral Health, and Northwestern University.

## Author Contributions

**Conceptualization:** Emily E. Haroz.

**Data curation:** Allison Ingalls, Fiona Grubin.

**Formal analysis:** Emily E. Haroz, Fiona Grubin.

**Investigation:** Allison Ingalls.

**Methodology:** Emily E. Haroz.

**Project administration:** Allison Ingalls.

**Supervision:** Allison Ingalls.

**Visualization:** Emily E. Haroz.

**Writing – original draft:** Emily E. Haroz, Allison Ingalls, Deborah Daro.

**Writing – review & editing:** Emily E. Haroz, Allison Ingalls, Karla Decker Sorby, Mary Dozier, Miranda P. Kaye, Michelle Sarche, Lauren H. Supplee, Daniel J. Whitaker, Fiona Grubin, Deborah Daro.

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
