## [Decision Letter · Decision Letter 0]

1 Jun 2022

PONE-D-21-34938Expert-generated standard practice elements for evidence-based home visiting programs using a modified Delphi processPLOS ONE

Dear Dr. Ingalls,

Thank you for submitting your manuscript to PLOS ONE. After careful consideration, we feel that it has merit but does not fully meet PLOS ONE’s publication criteria as it currently stands. Therefore, we invite you to submit a revised version of the manuscript that addresses the points raised during the review process.

 Please note that we have only been able to secure a single reviewer to assess your manuscript. We are issuing a decision on your manuscript at this point to prevent further delays in the evaluation of your manuscript. Please be aware that the editor who handles your revised manuscript might find it necessary to invite additional reviewers to assess this work once the revised manuscript is submitted. However, we will aim to proceed on the basis of this single review if possible. 

Your manuscript has been assessed by an expert reviewer, whose comments are appended to this letter. The reviewer has requested clarification on a few aspects of the reporting and made some suggestions for fruitful areas to enrich and expand your introduction and discussion to maximise the impact of your work. Please ensure you address each of the reviewer's points carefully in your response to reviewers document, and revise your manuscript accordingly.

We look forward to receiving your revised manuscript.

Kind regards,

Joseph Donlan

Editorial Office

PLOS ONE

Journal Requirements:

Reviewers' comments:

Reviewer's Responses to Questions

**Comments to the Author**

1. Is the manuscript technically sound, and do the data support the conclusions?

Reviewer #1: Yes

2. Has the statistical analysis been performed appropriately and rigorously? 

Reviewer #1: Yes

3. Have the authors made all data underlying the findings in their manuscript fully available?

Reviewer #1: Yes

4. Is the manuscript presented in an intelligible fashion and written in standard English?

Reviewer #1: Yes

5. Review Comments to the Author

Reviewer #1: This is a well-written and informative article describing the use of a modified Delphi process to generate expert opinions on standard practice elements (SPEs) and behavior change techniques necessary to achieve home visiting outcomes. Overall, I believe the article has much to offer the field of home visiting as explicit discussion of behavior change techniques tend to be limited. My biggest recommendation is to use the wealth of data collected to better chart a course for future work. The paragraph before Limitations gives several potential implications and future directions based on the study findings—the first suggesting that study results can “inform practice” by helping stakeholders personalize their programming. This seems to be one of the biggest future directions—i.e., how can home visiting models create behavior change frameworks that are personalized (keeping with the precision paradigm) and incorporate SPEs that align with behavior change frameworks. I suggest adding more details on exactly how this could be done by home visiting practitioners. Home visiting models that are considered evidenced-based may be reluctant to modify their approaches and the process of personalizing home visiting delivery seems like it would be a very involved process. I think this manuscript would have even greater impact if it expanded on this important point of informing practice and provided more details on what this process should look like, including stakeholders that need to be involved, how evidence-based models could be implemented using a personalized framework, and how home visitors would need to be trained to use a personalized approach. Without these additional details, I worry this article—while very informative and useful for the home visiting field—may not have the desired impact because it does not leave home visiting practitioners with enough food for thought on how their models/programs need to be modified. Other specific comments are noted below:

The Introduction is well-written and gives the reader a good sense of the landscape of evidence-based home visiting (EBHV) programs. I do think the comment on lines 80-84 about the challenges in identifying and selecting programs could benefit from slight rewording. My recollection is that the HV evidence review (HOMEVEE) provides a snapshot of each EBHV model, with details on populations served, evidence related to various maternal and child health outcomes, etc. The authors should note this and then perhaps give a bit more explanation on why that level of detail is still insufficient/confusing to stakeholders looking to select a model.

Lines 92-94: Can you indicate who is encouraging program developers to “unravel their approach and identify with greater specificity their key design elements…” Is this something that is being encouraged by MIECHV, researchers, policy-makers, or some combination?

The top of Figure 1 is blurry and hard to read.

For readers unfamiliar with Delphi and group consensus techniques, it might be helpful to add another sentence or two when Delphi is first introduced to give a sense of its key characteristics and how it is typically used. A reference would also be helpful.

Related point—in the Procedures section there is reference to modifying the Delphi process, but I don’t think the reader has enough information on what a “standard” Delphi looks like to be able to understand the modifications made.

The last part of the Introduction after the paper’s goals are introduced is a little hard to follow. The second goal as written is quite long and there appears to be a typo in the last sentence (“because of the overlapping but funding…”).

It is reported that 16 individuals comprised the expert panel, but when describing them it appears 10 were researchers, 7 model reps, and 7 tribal stakeholders. I’m assuming that the three categories overlapped—i.e., someone could be part of two groups—but that should be clarified so that these n’s are aligned.

As I understand the way the Delphi process took place, respondents were asked to generate lists of behavior change techniques necessary for achieving various maternal and child health outcomes relevant to home visiting. What seems to be missing in this approach, however, is understanding how well/to what extent existing evidence-based models incorporate these behavior change technique since it doesn’t appear respondents were asked to use that frame in their responses. I think this is an important limitation and future direction that needs to be addressed.

6. PLOS authors have the option to publish the peer review history of their article (what does this mean?). If published, this will include your full peer review and any attached files.

Reviewer #1: No

---

## [Author Response · Author response to Decision Letter 0]

30 Aug 2022

Please see attached file with full response to reviewers (Word file name Response to Reviewers)

---

## [Decision Letter · Decision Letter 1]

27 Sep 2022

Expert-generated standard practice elements for evidence-based home visiting programs using a Delphi process

PONE-D-21-34938R1

Dear Dr. Ingalls,

We’re pleased to inform you that your manuscript has been judged scientifically suitable for publication and will be formally accepted for publication once it meets all outstanding technical requirements.

Kind regards,

Tanya Doherty, PhD

Academic Editor

PLOS ONE

Additional Editor Comments (optional):

Reviewers' comments:

Reviewer's Responses to Questions

**Comments to the Author**

1. If the authors have adequately addressed your comments raised in a previous round of review and you feel that this manuscript is now acceptable for publication, you may indicate that here to bypass the “Comments to the Author” section, enter your conflict of interest statement in the “Confidential to Editor” section, and submit your "Accept" recommendation.

Reviewer #1: All comments have been addressed

2. Is the manuscript technically sound, and do the data support the conclusions?

Reviewer #1: Yes

3. Has the statistical analysis been performed appropriately and rigorously? 

Reviewer #1: Yes

4. Have the authors made all data underlying the findings in their manuscript fully available?

Reviewer #1: Yes

5. Is the manuscript presented in an intelligible fashion and written in standard English?

Reviewer #1: Yes

6. Review Comments to the Author

Reviewer #1: (No Response)

7. PLOS authors have the option to publish the peer review history of their article (what does this mean?). If published, this will include your full peer review and any attached files.

Reviewer #1: No

---

## [Editor Report · Acceptance letter]

6 Oct 2022

PONE-D-21-34938R1 

Expert-generated standard practice elements for evidence-based home visiting programs using a Delphi process 

Dear Dr. Ingalls:

I'm pleased to inform you that your manuscript has been deemed suitable for publication in PLOS ONE. Congratulations! Your manuscript is now with our production department. 

Kind regards, 

on behalf of

Professor Tanya Doherty 

Academic Editor

PLOS ONE